EMBO
Molecular Medicine

# Sequence variation in *PPP1R13L* results in a novel form of cardio-cutaneous syndrome

Tzipora C Falik-Zaccai[1,2,*] [ID], Yiftah Barsheshet[2,†] [ID], Hanna Mandel[3,4,†], Meital Segev[2], Avraham Lorber[4,5], Shachaf Gelberg[2], Limor Kalfon[1], Shani Ben Haroush[1], Adel Shalata[6] [ID], Liat Gelernter-Yaniv[7], Sarah Chaim[1], Dorith Raviv Shay[1], Morad Khayat[8], Michal Werbner[2], Inbar Levi[1], Yishay Shoval[1], Galit Tal[3,4], Stavit Shalev[4,8], Eli Reuveni[2], Emily Avitan-Hersh[9], Eugene Vlodavsky[4,10], Liat Appl-Sarid[11], Dorit Goldsher[4,12], Reuven Bergman[4,9], Zvi Segal[2,13], Ora Bitterman-Deutsch[2,14] & Orly Avni[2,**] [ID]

## Abstract

**Dilated cardiomyopathy (DCM) is a life-threatening disorder whose genetic basis is heterogeneous and mostly unknown. Five Arab Christian infants, aged 4–30 months from four families, were diagnosed with DCM associated with mild skin, teeth, and hair abnormalities. All passed away before age 3. A homozygous sequence variation creating a premature stop codon at *PPP1R13L* encoding the iASPP protein was identified in three infants and in the mother of the other two. Patients' fibroblasts and *PPP1R13L*-knocked down human fibroblasts presented higher expression levels of pro-inflammatory cytokine genes in response to lipopolysaccharide, as well as *Ppp1r13l*-knocked down murine cardiomyocytes and hearts of *Ppp1r13l*-deficient mice. The hypersensitivity to lipopolysaccharide was NF-κB-dependent, and its inducible binding activity to promoters of pro-inflammatory cytokine genes was elevated in patients' fibroblasts. RNA sequencing of *Ppp1r13l*-knocked down murine cardiomyocytes and of hearts derived from different stages of DCM development in *Ppp1r13l*-deficient mice revealed the crucial role of iASPP in dampening cardiac inflammatory response. Our results determined *PPP1R13L* as the gene underlying a novel autosomal-recessive cardio-cutaneous syndrome in humans and strongly suggest that the fatal DCM during infancy is a consequence of failure to regulate transcriptional pathways necessary for tuning cardiac threshold response to common inflammatory stressors.**

**Keywords** dilated cardiomyopathy; genetics; inflammation; myocarditis; *PPP1R13L*
**Subject Categories** Cardiovascular System; Genetics, Gene Therapy & Genetic Disease; Immunology

## Introduction

Dilated cardiomyopathy (DCM) is characterized by impaired contraction of heart ventricles leading to overt heart failure. It results in ~10,000 deaths per year in the United States and is the primary indication for cardiac transplantation. Most patients present symptoms during adulthood, but DCM may also occur in childhood. Genetically inherited DCM is highly diverse, caused by over 40 genes (Kloos *et al*, 2012) and mostly inherited in an autosomal-dominant manner. Numerous syndromes, many with genetic etiologies, have been reported in association with DCM. Syndromic DCM may involve skin, CNS, and musculo-skeletal systems. Inherited cardio-cutaneous syndromes (CCSs) such as Naxos and Carvajal are two examples, caused by genes encoding desmosomal proteins. These CCSs are clinically characterized by woolly hair, palmoplantar keratoderma, and skin fragility, all of which appear in early

1  Institute of Human Genetics, Galilee Medical Center, Nahariya, Israel
2  Faculty of Medicine in the Galilee, Bar-Ilan University, Safed, Israel
3  Metabolic Disease Unit, Rambam Health Care Campus, Haifa, Israel
4  Rappaport Faculty of Medicine, Technion, Israel Institute of Technology, Haifa, Israel
5  Department of Pediatric Cardiology, Rambam Health Care Campus, Haifa, Israel
6  The Winter Genetic Institute, Bnei Zion Medical Center, Haifa, Israel
7  Pediatric Cardiology Clinic, Bnei Zion Medical Center, Haifa, Israel
8  The Genetic Institute, Ha'emek Medical Center, Afula, Israel
9  Department of Dermatology, Rambam Health Care Campus, Haifa, Israel
10 Department of Pathology, Rambam Health Care Campus, Haifa, Israel
11 Department of Pathology, Galilee Medical Center, Nahariya, Israel
12 Department of Diagnostic Imaging, Rambam Health Care Campus, Haifa, Israel
13 Department of Ophthalmology, Galilee Medical Center, Nahariya, Israel
14 Dermatology Clinic, Galilee Medical Center, Nahariya, Israel
   *Corresponding author. Tel: +972 4 9107493; E-mail: falikmd.genetics@gmail.com
   **Corresponding author. Tel: +972 72 2644921; E-mail: orly.avni@biu.ac.il
   †These authors contributed equally to this work

childhood, and cardiac abnormality manifested from young adulthood (Bolling & Jonkman, 2009; Jiang *et al*, 2011; Rickelt & Pieperhoff, 2012).

iASPP (the inhibitor of apoptosis-stimulating protein of p53) is an evolutionarily conserved member from worm to human of the ASPP (ankyrin repeat, SH3 domain, and proline-rich region containing protein) family (Yang *et al*, 1999; Bergamaschi *et al*, 2003; Herron *et al*, 2005; Minekawa *et al*, 2007; Chikh *et al*, 2011; Toonen *et al*, 2012). iASPP is encoded by *PPP1R13L* and expressed mainly in epithelial cells of the skin, testes, heart, and stomach, but also in other tissues. iASPP is associated with multiple proteins including the transcription factors NF-κB (Yang *et al*, 1999; Herron *et al*, 2005) and p53 (Bergamaschi *et al*, 2003, 2006). Functionally, iASPP inhibits transcriptional activity of NF-κB and p53, though the underlying mechanism is unclear.

We report for the first time a novel human CCS, associated with a premature stop codon in *PPP1R13L* and ablation of iASPP expression in three Arab Christian infants and in the mother of two additional infants. Our data show how the loss of iASPP lowers the heart's threshold to inflammatory response and strongly suggests that this hypersensitivity underlies the severe DCM seen in our patients.

# Results

### Patients and families

Clinical, dermatologic, and cardiologic details of five Arab Christian patients (Fig 1A) aged 4–30 months, diagnosed with autosomal-recessive (AR) CCS (Fig 1B and C) following an inter-current viral infection, are summarized in Table 1. Parents of all patients are consanguineous and reside in the same village. Three patients presented with unusually sparse and woolly hair and two had wedged teeth and dry skin (Fig 1B). Dermatologic evaluation was

unavailable for patients $VI_{12}$ and $VI_{13}$. Patient $VI_4$ presented with bilateral cloudy cornea and congenital corneal cyst, and no behavioral visual response.

Metabolic analyses and skeletal muscle biopsy were normal. All patients passed away before reaching 3 years of age. Post-mortem histology of one heart (patient $VI_{10}$) revealed myocarditis (Fig 1D). The parents of the other patients refused post-mortem analysis. Couple $V_3$ and $V_4$ electively aborted four fetuses between weeks 21 and 31 of gestation with midline brain, eye, and facial defects (patients $VI_{5-8}$). In the extended family, two more affected girls ($VI_{12}$, $VI_{13}$) passed away before initiation of this study and no DNA/fibroblasts were available from them. No pathologic sequence variations (SeVa) were found in the five main genes involved in CCS (TP63 (MIM 603273), EDAR (MIM 604095), DSP (MIM 125647), PKP2 (MIM 604536), and JUP (MIM 173325)) sequenced in $VI_{10}$.

### Homozygosity mapping, haplotype analysis, and whole exome deep sequencing (WES)

Homozygosity SNP analysis revealed a large homozygous region on chromosome 19 (chr19: 44615969-55644865). WES of $VI_{10}$, validated by Sanger sequencing, identified a homozygous SeVa, which creates a premature stop codon in exon 11 (c.2241C> G, p.Tyr747Ter) of *PPP1R13L* (Fig 1E). *PPP1R13L* is located on chromosome 19q13.1 (chr19: 45897911-45909607; MIM 607463) and encodes the iASPP protein. Segregation analyses (Fig 1F) confirmed homozygosity of this SeVa in three patients and carrier state in their parents. The mother ($V_6$) of the affected sisters $VI_{12}$ and $VI_{13}$ (from whom DNA was not available) was found to be a carrier (Fig 1A), as well as the maternal grandmother ($IV_8$). The father-$V_5$ presented with sudden death and was unavailable for testing. Four fetuses ($VI_{5-8}$) were also homozygous for this SeVa. Most likely, in this highly inbred family, the two different disorders, CCS and facial/brain malformations of the fetuses, were

---

**Figure 1.  Clinical characterization of the CCS patients.**

A   Pedigree of the extended family. Filled symbols indicate affected members. Arrow indicates the proband. Circles, females; squares, males; slant, deceased; and asterisk, no DNA available for molecular diagnosis.

B   Phenotype of the patients as indicated. (a) Sparse woolly hair. (b) Protrusion of upper lip due to deformity of teeth. (c) Ichthyosis-like fine scale and erythema.

C   Images of the hearts of the patients as indicated. (a) M-mode echocardiography taken from the parasternal long-axis view of the left ventricle (LV) illustrating poor LV function with severe involvement of the interventricular septum (IVS) and reduced motion of the left ventricular posterior wall (LVPW). (b) Bi-dimensional apical four-chamber view with swirling of SMOG heart chambers as a result of poor myocardial function. The SMOG is enhanced in the right atrium. (c) Bi-dimensional apical four-chamber view with free tricuspid valve regurgitation (TR) due to lack of cooptation of the tricuspid valve leaflets during systole (marked by a red bar) as a result of critical reduction of right ventricular function and right ventricular enlargement. (d) Right and left ventricular thrombi seen in the short-axis apical parasternal view. T1, large right ventricular thrombus; T2 and T3, left ventricular thrombi. (e) The color-flow Doppler picture: TR. Free regurgitation of the tricuspid valve shown in modified four chambers apical view. RA, right atrium; RV, right ventricle; LA, left atrium; LV, left ventricle; and TR, tricuspid regurgitation color-flow Doppler. (f) M-mode from long-axis view of the left ventricle; moderately reduced LV function with fractional shortening of 26%, enlarged RV. (g) Modified four-chamber view: severe right atrium enlargement, no cooptation of the tricuspid valve leaflets during systole, leading to severe TR. (h) Modified four-chamber view: severe right atrium enlargement, severe tricuspid regurgitation.

D   Photomicrograph demonstrating fibrosis and interstitial inflammation in the heart of patient $VI_{10}$. (a, c) Post-mortem heart sections (2.7 Y). (a) Prominent fibrosis and lymphocytic interstitial inflammation (hematoxylin & eosin ×50). (c) Subepicardial inflammation and mild interstitial "myocarditis" (hematoxylin & eosin ×100). (b, d) Fetal heart ($VI_7$) homozygous for p.Tyr747Ter. No inflammatory infiltrate or other histological abnormalities are seen. (b) Normal arrayed myocardium (hematoxylin & eosin ×50). (d) Subepicardial region with no histopathological changes (hematoxylin & eosin ×100).

E   Sanger sequencing confirming that the patient $VI_{10}$ is homozygous for the causative SV c.2241C > G, p.Tyr747Ter in *PPP1R13L*.

F   Segregation of c.2241C > G, p.Y747X in *PPP1R13L* in the studied family. The affected girl $VI_{10}$ is homozygous for the causative SV p.Y747X, as are the four fetuses that were aborted. The parents are obligate carriers as shown, and, of the two healthy sisters, one carries the wild-type allele only, and the other is heterozygous for the causative SV.

Source data are available online for this figure.

caused by mutations in distinct genes. Alternatively, the facial/brain malformations might have an incomplete penetrance with this CCS. Haplotype analysis using DNA markers closely linked to

*PPP1R13L* D19S903, D19S902 support a common founder haplotype in all patients (data not shown). The c.2241C>G in *PPP1R13L* is not present in dbSNP or in the 1000 Genomes project databases.

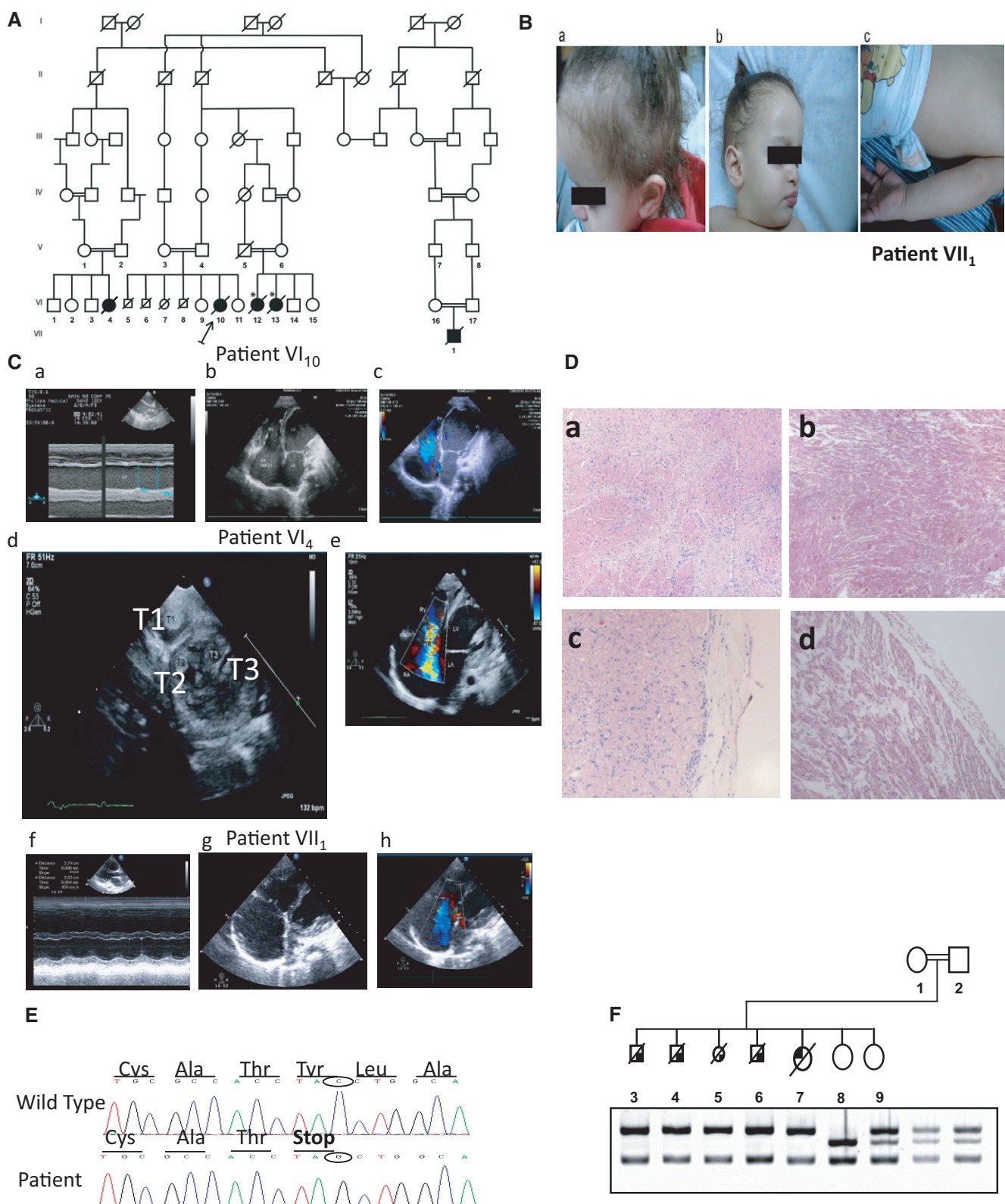

**Figure 1.**

**Table 1. Clinical characteristics of the five patients reported in this study with a novel CCS.**

| | Patient M.A (VI$_{10}$) | Patient A.J (VII$_1$) | Patient Y.M (VI$_4$) | R (VI$_{12}$) older sister | R (VI$_{13}$) |
|---|---|---|---|---|---|
| DCM Age of presentation/ diagnosis (months) | 6 | 12 | 30 | 4 | 10 |
| Consanguinity | + | + | + | + | + |
| Age at death | 2.8 years | 1.5 years | 2.8 years | 9 months | 1.5 years |
| **Cardiac** | | | | | |
| Severe DCM | + | + | + | + | + |
| Mitral valve regurgitation | ++ | ++ | + | | |
| Severe tricuspid valve regurgitation | + | + | + | | |
| Small pericardial effusion | + | | ++ | | |
| On EKG: multiple bi-geminis, multiple VPB's[a] | + | | + | | Sinus tachycardia + |
| Disarray and vacuolization of cardiomyocytes | + | NA | NA | NA | NA |
| Bicuspid aortic valve | | + | | | |
| Dominant RV | − | − | + Severely enlarged RV with a severe reduction in RV contraction Severely enlarged right atrium Moderately reduced LV function | − | − |
| Dominant LV | + Left ventricular enlargement with reduced systolic function with marked involvement of the interventricular septum followed by biventricular dysfunction and akinetic RV and tricuspid regurgitation | + Left ventricular hypertrophy and dilation with marked reduction of LV systolic function followed by biventricular dysfunction with RV failure and tricuspid regurgitation | − | + VSD, marked reduced LV function enlarged heart shadow on chest X-ray | + VSD, marked reduced LV function enlarged heart shadow on chest X-ray |
| **Skin** | | | | | |
| Impaired thermoregulation | − | − | − | NA | NA |
| Sparse, dry, wooly hair | + | + | + | NA | NA |
| Ichthyosis-like fine scale and erythema | + | − | + | NA | NA |
| Pruritus | − | − | + | NA | NA |
| Fibroma | − | − | + | NA | NA |
| Palmo-plantar keratoderma | − | − | − | NA | NA |
| Dystrophic nails | + | − | − | NA | NA |
| Wedge-shaped teeth | + | + | + | NA | NA |

 

**Table 1**  (continued)

| | Patient M.A (VI$_{10}$) | Patient A.J (VII$_1$) | Patient Y.M (VI$_4$) | R (VI$_{12}$) older sister | R (VI$_{13}$) |
|---|---|---|---|---|---|
| Orthopedic | | | | | |
| Intoeing bilateral | + | + | + | NA | NA |
| Facial cysmorphism | | | | | |
| High forehead hairline | + | + | + | NA | NA |
| Border/low and depressed nasal bridge | + | + | + | NA | NA |
| Cleft lip and palate | − | − | − | + | + |
| Ophthalmological abnormalities | − | − | Bilateral cloudy cornea and congenital corneal cyst No behavioral visual response | − | − |
| Failure to thrive | + | + | − | + | + |
| Motor development | Delayed | Delayed | Delayed | NA | NA |
| Cognitive development | Normal | Normal | Normal | NA | NA |
| Fetal brain U/S | Normal | Normal | Normal | | |
| Brain MRI | NA | NA | Distortion of the eyeball with sand clock appearance, thin optic nerves, cystic protrusion of the eyeball LT> RT Decreased brain volume secondary ventriculomegaly. Decreased white matter maturity for age. | NA | NA |
| Brain CT | | | Middle cerebral artery infarct | | |

VSD, ventricular septal defect; LV, left ventricle; RV, right ventricle.
Data were collected regarding medical history, metabolic measurements, imaging, electrophysiological studies, and muscle biopsy. Complete physical, dermatological, and cardiological examinations were performed on the five patients. The disease phenotype in all patients is similar.
[a]Ventricular Premature Beats.

To identify the SeVa frequency among the Arab Christian community who reside in the same village and among the Arab Christian community in northern Israel, one hundred individuals from each group were studied. Eleven heterozygous carriers were identified in the first group with no healthy individuals found to be homozygous to this SeVa. One heterozygous carrier was identified in the second group, thus indicating existence of a genetic isolate for this novel CCS.

### *PPP1R13L* SeVa and expression

*PPP1R13L* mRNA level was lower in the VI$_4$, VI$_{10}$, and VII$_1$ patient skin-derived fibroblasts (PDFs) than in control age-matched skin-derived fibroblasts (CFs; Fig 2A). The iASPP protein was completely absent in PDFs of VI$_4$, VI$_{10}$, and VII$_1$ (Fig 2B and C). *PPP1R13L* knockdown confirmed that the band appearing specifically in the CFs and not in the PDFs was iASPP (Fig 2D).

### NF-κB-dependent pro-inflammatory response

Considering that iASPP interacts and represses transcriptional activity of NF-κB (Yang *et al*, 1999; Takada *et al*, 2002; Herron *et al*, 2005; Sullivan & Lu, 2007) and that dysregulation of NF-κB is involved in the development of heart failure (Gordon *et al*, 2011), we hypothesized that absence of iASPP can unleash NF-κB to increase transcription of pro-inflammatory mediators and, as a consequence, induce prolonged inflammatory processes and

eventually DCM. In accordance with our hypothesis, PDFs from either one of the two patients VI$_{10}$ and VII$_1$ expressed higher mRNA levels of the inflammatory cytokines *IL1B, TNFA, IL6,* and of the chemokine *IL8* than CFs following stimulation with LPS (Fig 3A and B). The mRNA expression level of another known NF-κB-target gene, the chemokine *MCP1*, increased similarly in PDFs and CFs. Two other putative NF-κB-target genes *MMP9* and *IKB* were induced neither in the CFs nor in PDFs. These results may suggest a restricted effect of iASPP on expression of selected NF-κB-target genes. In VI$_4$-PDFs, we determined increased expression of *IL1B* and *TNFA* but not of *IL6* and *IL8* (data not shown). This may reflect heterogeneity in human fibroblasts. To further dissect the effect of iASPP deficiency on the expression of pro-inflammatory mediators, we knocked *PPP1R13L* down in CFs (Fig 3C). The mRNA levels of the inflammatory mediators were higher in response to LPS stimulation in the *PPP1R13L*-silent CFs than in control CFs, confirming the role of iASPP in the regulation of the inflammatory response.

Amount of nuclear NF-κB subunit p65 was higher and its nuclear duration longer following LPS stimulation in VI$_{10}$-PDFs than in CFs (Fig 4A). ChIP assay demonstrated that p65 associated more strongly with *IL1B* promoter 1 h following LPS stimulation in PDFs than in CFs, where binding activity was almost below detection (Fig 4B). Similar results were obtained using primers amplifying the *TNFA* promoter. In contrast, binding activity of p65 was induced neither at the p53-target gene *p21* promoter in both PDFs and CFs nor at the *IKB* promoter, whose expression was unresponsive to the presence of LPS or iASPP under these experimental conditions. To

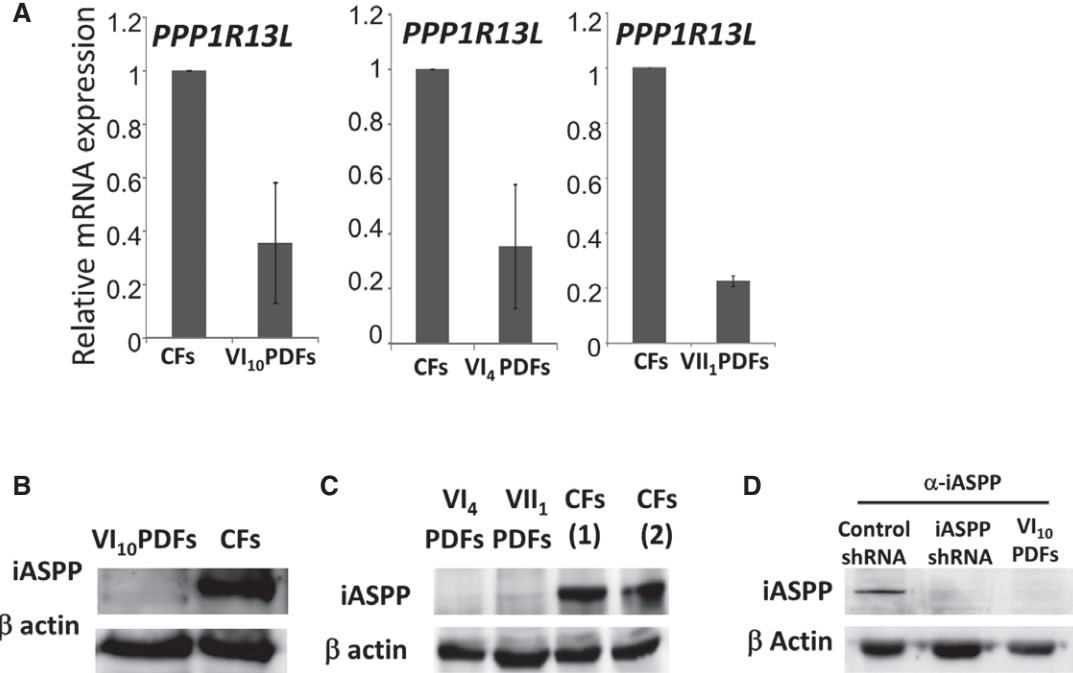

**Figure 2. *PPP1R13L* SeVa and expression.**

A    *PPP1R13L* mRNA expression level is lower in $VI_{10}$ PDFs than in CFs (set as 1). The qPCR results are the mean values of three independent experiments $\pm$ SD.

B, C   Western blot (WB) analyses demonstrated the absence of iASPP in $VI_{10}$, $VI_4$, and $VII_1$ PDFs.

D    WB analysis demonstrated that *PPP1R13L*-shRNA downregulated the putative iASPP band.

Source data are available online for this figure.

finally confirm involvement of NF-κB downstream to iASPP in the regulation of inflammatory genes, $VI_{10}$-PDFs and CFs were stimulated in the presence or absence of the pharmacologic inhibitor of NF-κB activity BOT-64. As seen in Fig 4C, the induced expression of the inflammatory cytokine genes *IL1B*, *IL6*, and *TNFA* in the PDFs as well as in the CFs was almost completely NF-κB-dependent. All together, these results indicate an elevated activity of NF-κB in the absence of iASPP.

### *Ppp1r13l*-knocked down cardiomyocytes are hypersensitive to inflammatory stimulus

Next, we knocked *Ppp1r13l* down in newborn murine cardiomyocytes (Fig 5A). As in human fibroblasts, LPS-inducible expression of the pro-inflammatory cytokine genes *Il1b*, *Tnfa* and *Il6* was higher in the silent cells than in controls (Fig 5B). To reveal genome-wide expression patterns of iASPP-regulated genes in cardiomyocytes, we performed RNA-seq. First, we compared expression levels without any stimulation and found that several genes with potential to promote inflammatory condition, as those involved in cellular adhesion and antigen presentation (major histocompatibility complex (MHC) class I), were already upregulated without any further intervention (Appendix Fig S1). Also, mRNA levels of several known DCM-associated genes, which are required for normal cardiac function, were reduced (Appendix Fig S1). As expected, following treatment with LPS for 4 h, a cohort of genes encoding pro-inflammatory mediators as chemokines, cytokines, matrix metalloproteinases, and

nitric oxide synthase were upregulated more strongly in the *Ppp1r13l*-knocked down cells (Appendix Fig S1). Considering that abnormal function of the innate immune response is associated with heart failure and DCM (Cao *et al*, 2011; Gullestad *et al*, 2012; Takahashi, 2014; Butts *et al*, 2015; Prabhu & Frangogiannis, 2016), these results suggest that a pro-inflammatory transcriptional pattern is underlying the disease initiation/propagation and consequently the adverse cardiac remodeling in our patients.

### Dynamic development of inflammatory transcriptional patterns in *Ppp1r13l*-deficient hearts

To study *in vivo* the cardiac transcriptional programs in the absence of iASPP, we used the wa3 mice that carry spontaneous AR mutation in *Ppp1r13l*. These mice possess similar phenotype to the CCS patients with wavy hair and fatal post-natal progressive DCM (Herron *et al*, 2005; Toonen *et al*, 2012; Appendix Fig S2). Mutation in *Ppp1r13l* in mice also causes open eyelids at birth that culminating in blind adults (Toonen *et al*, 2012; Appendix Fig S2). As can be seen in Fig 6A, the mRNA of the pro-inflammatory cytokine *Il1b* was expressed at much higher level in the hearts of 12-week-old wa3 mice than in littermate control WT mice. The increased expression of *Il1b* without experimental exposure to external inflammatory stimuli may reflect either a cardiac intrinsic loose regulation of the cytokine transcription, or, alternatively, a hyper-response to internal cues as commensal microbiota or mechanical stress, which normally do not exceed the threshold for activation of the innate

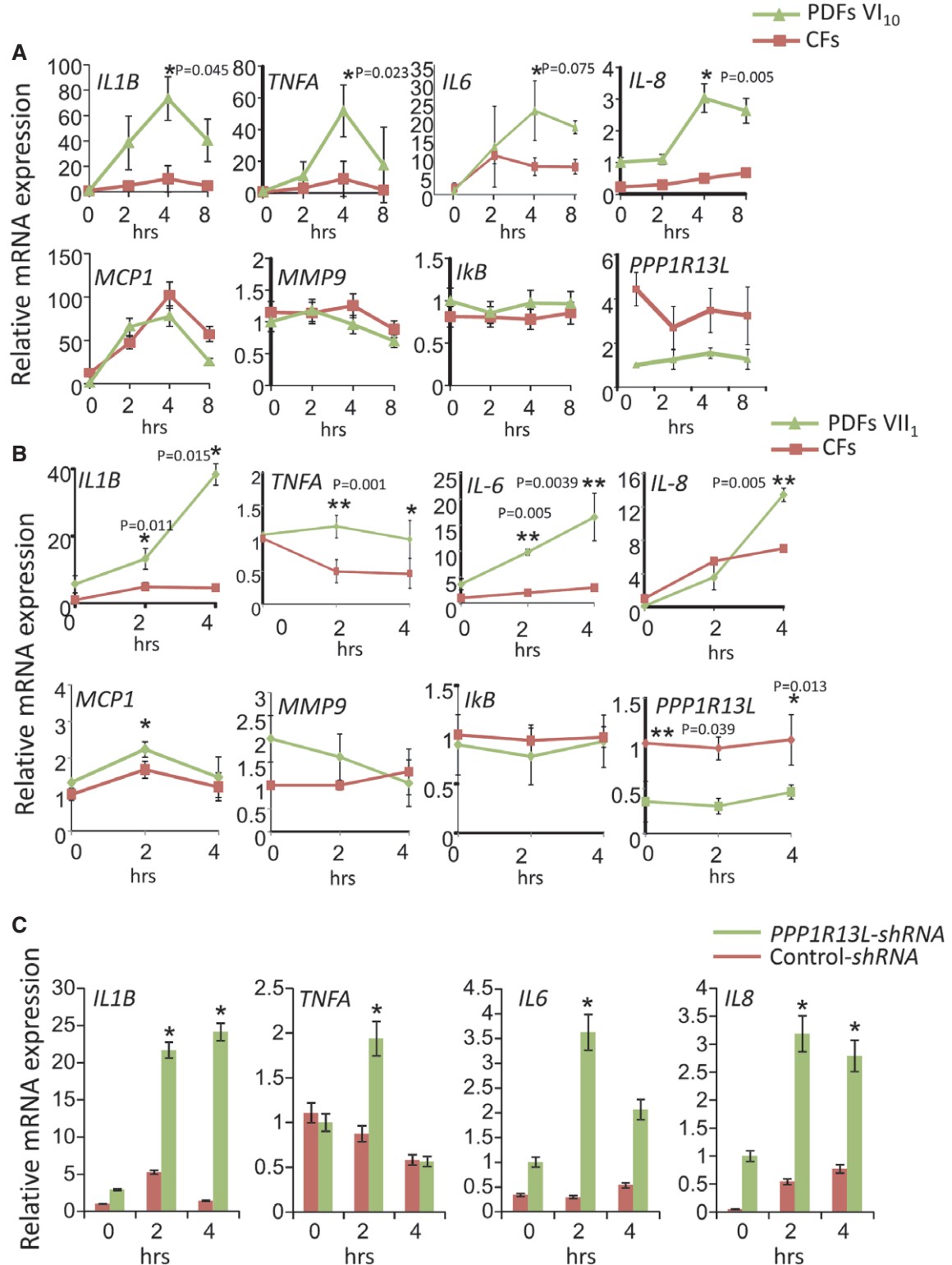

**Figure 3.  Upregulation of pro-inflammatory mediators in PDFs.**

A, B   Higher induction of inflammatory cytokine mRNA levels in LPS-stimulated VI$_{10}$ PDFs (A) or VII$_{1}$ PDFs (B) than in CFs. The expression level in resting conditions of the PDFs was set as 1. The qPCR results are the mean values of at least three independent experiments ± SD. Differences between knockdown and control with *P*-values ≤ 0.05 (Student's *t*-test) are indicated with an asterisk.

C   Knockdown of *PPP1R13L* in CFs resulted in increased mRNA expression levels of inflammatory mediators. The qPCR results are the mean values of three independent experiments ± SD. Differences between knockdown and control with *P*-values (Student's *t*-test) are indicated with asterisk in the graph.

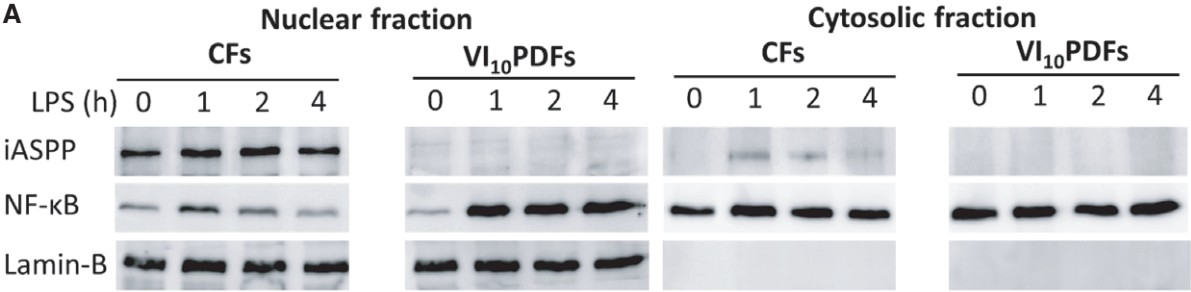

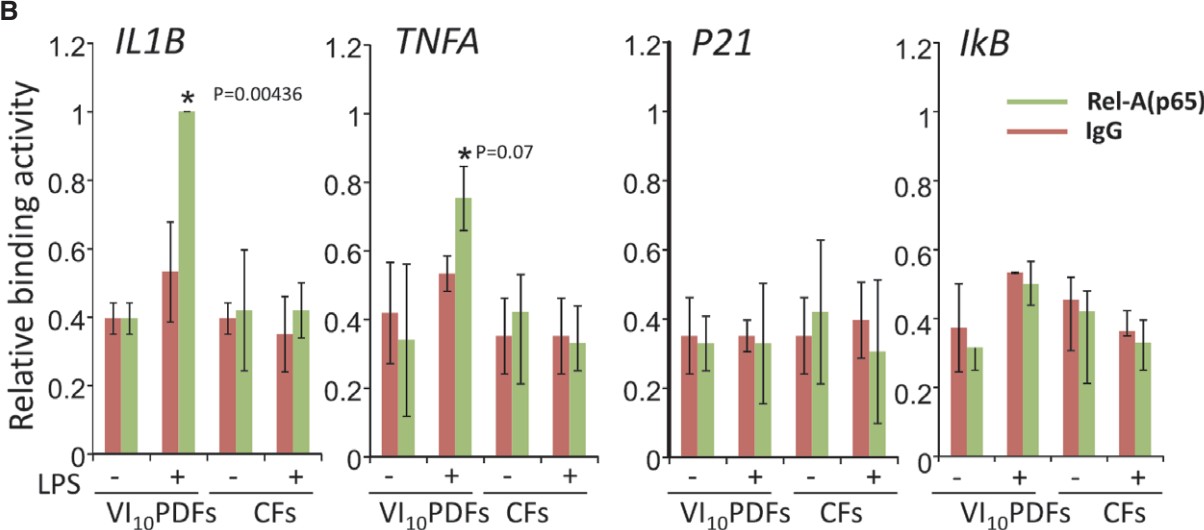

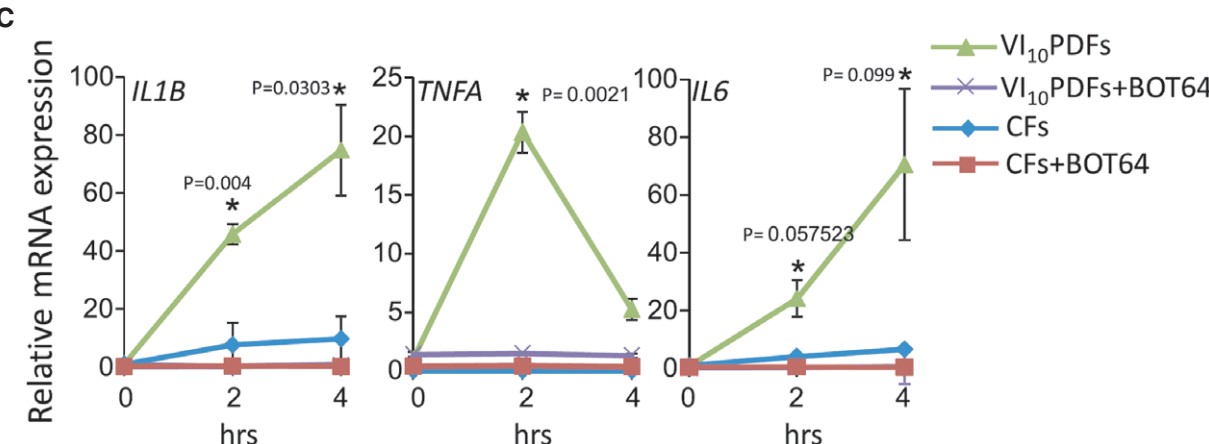

**Figure 4.  NF-κB-dependent upregulation of pro-inflammatory cytokines.**

A   WB analysis of nuclear and cytosolic fractions demonstrated that the duration of p65 in the nucleus is longer in the VI$_{10}$ PDFs than in CFs.

B   ChIP assay shows that p65 is associated more strongly with inflammatory cytokine gene promoters in LPS-stimulated PDFs than in LPS-stimulated CFs. The binding activity at the *IL1B* promoter was set as 1. Differences between knockdown and control with *P*-values (Student's *t*-test) are indicated with an asterisk and exact values.

C   NF-κB inhibitor BOT-64 blocked the inducible mRNA expression of the inflammatory cytokines. The qPCR results are the mean values of three independent experiments ± SD. Differences between knockdown and control with *P*-values (Student's *t*-test) are indicated with an asterisks and exact values.

Source data are available online for this figure.

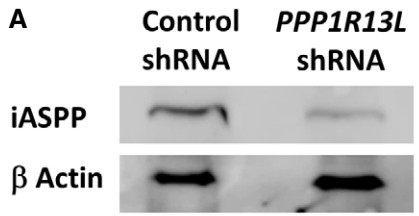

**A**

Control shRNA    PPP1R13L shRNA

iASPP

β Actin

**B**

PPP1R13L-shRNA1
Control-shRNA

*Il1b*

Relative mRNA expression

P=0.004 ***

*Tnfa*

P=0.001 ***

*Il6*

P=0.0001 ***

PPP1R13L-shRNA2
Control-shRNA

*Il1b*

Relative mRNA expression

P=0.012 *

*Tnfa*

P=0.008 *

**Figure 5.  iASPP regulates the innate immune response of murine neonatal cardiomyocytes.**

A   WB analysis confirmed the reduced expression of iASPP in *Ppp1r13l*-knocked down cardiomyocytes.

B   qPCR showed that *Ppp1r13l*-knocked down murine cardiomyocytes (using two alternative shRNA sequences) expressed higher level of pro-inflammatory cytokine mRNAs after 4 h of LPS stimulation. The expression in *Ppp1r13l*-knocked down murine cardiomyocytes was set as 1. The qPCR results are the mean values of three independent experiments ± SD. Genome-wide expression analysis is presented in Appendix Fig S1. Differences between knockdown and control with *P*-values (Student's *t*-test) are indicated with asterisks and exact values.

immune response of the heart. The expression of selected regulatory and structural genes was not significantly differed (Fig 6A), suggesting that the increased expression of inflammatory mediators precedes alteration in the expression of genes required for normal cardiac function. To widely assess dynamic changes in transcriptional patterns during the DCM development, we employed RNA-seq for newborn, 7-week-old and 12-week-old wa3- and littermate-derived hearts. Hearts of newborn wa3 mice expressed differentially

in a significant ("R" Deseq $P < 0.05$ threshold) manner only one mRNA of the *s100a9* gene (Fig 6B). The S100A9 (calgranulin B) and its binding partner S100A8 are expressed primarily by myeloid cells, and their expression is inducible by inflammatory stimuli in endothelial cells and cardiomyocytes (Averill *et al*, 2012). Seven weeks later, 1,050 genes were differentially expressed ($P < 0.05$ threshold Deseq algorithm "R"; Fig 6B and C, and list in Dataset EV1). 269 genes were 1.5- to ~40-fold upregulated. Genes that their

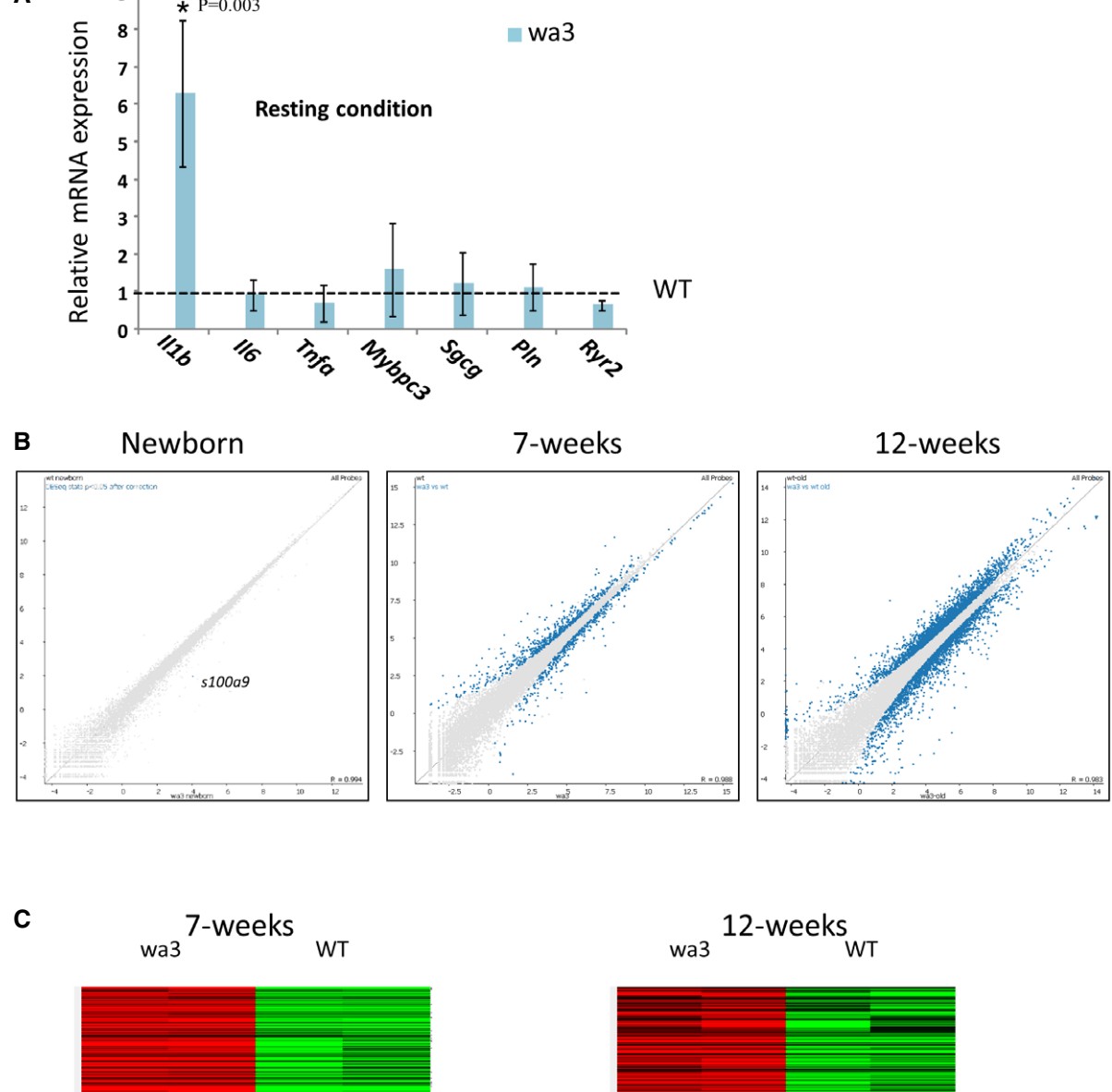

**Figure 6.  Hyper cardiac inflammatory response of *Ppp1r13l*-deficient mice.**

A   wa3 hearts express higher level of *Il1* than littermate WT hearts without significant changes in selected structural and regulatory genes as myosin binding protein C (*Mybpc3*), phospholamban (*Pln*), ryanodine receptor 2 (*Ryr2*), and sarcoglycan gamma (*Sgcg*). Hearts were removed from perfused mice, and mRNA was subjected for qPCR. The qPCR results are mean values of three independent experiments ± SD. Difference between knockdown and control with *P*-value = 0.003 (Student's *t*-test) is indicated with an asterisk.

B   A scatter plot showing the deviation from normal expression by highlighting the differentially expressed genes of newborn, 7-week- and 12-week-old hearts (*P* < 0.05 threshold, calculated by Deseq algorithm "R"). The expression level of *s100a9* was higher in wa3 mice than in WT by 2.8 times. For the full list of genes, see Datasets EV1 and EV2, and functional analyses in Appendix Figs S3–S6.

C   Hierarchical clustering heat map of differentially expressed genes in the hearts of 7- and 12-week-old wa3 mice in comparison with WT.

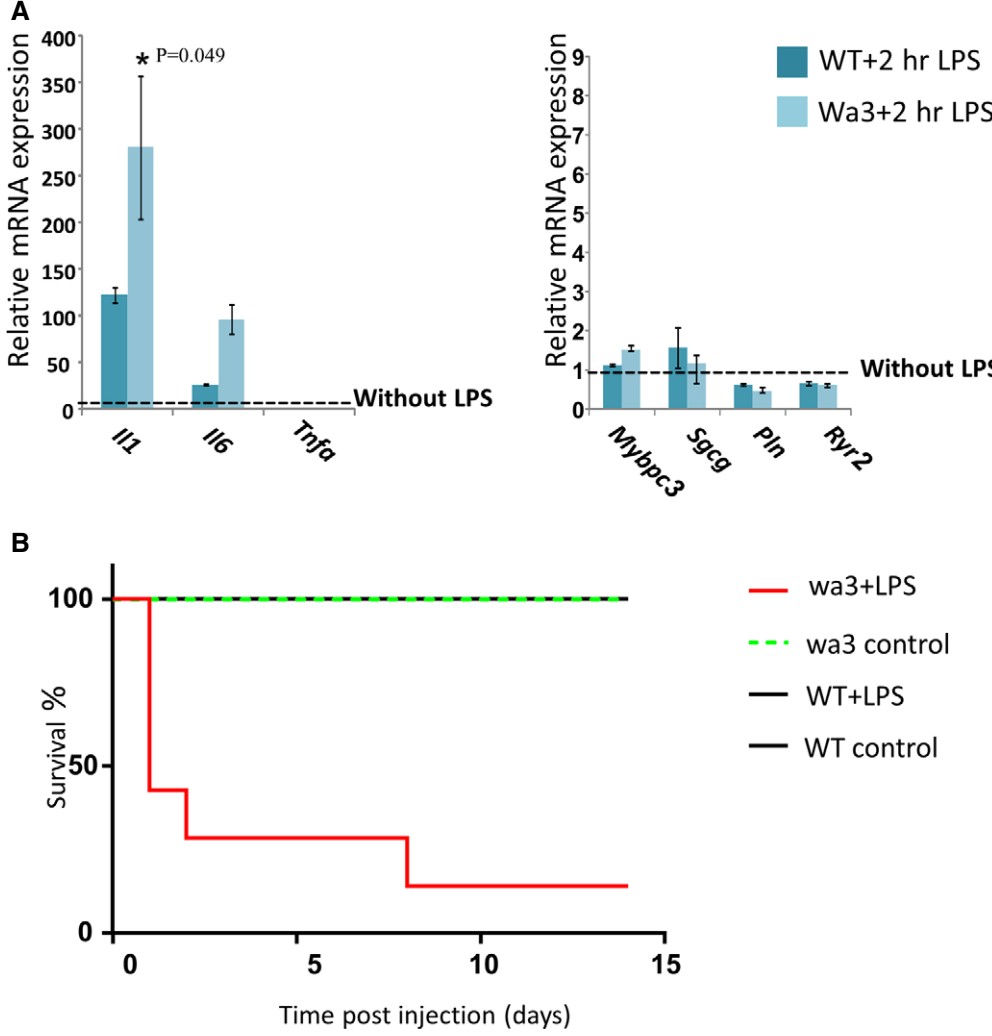

**Figure 7. *Ppp1r13l*-knockout mice exhibits cardiac hypersensitivity to inflammatory triggers.**

A   LPS (5 mg/kg for 2 h) induced higher expression levels of cytokine genes in wa3 hearts than in littermate WT hearts. The results are presented as fold induction of untreated controls. The mRNA expression levels of *Pln* and *Ryr2* were reduced in both wa3 and WT hearts, while the expression levels of *Mybpc3* and *Sgcg* were unchanged. The qPCR results are the mean values of three independent experiments ± SD. For the full list, see Dataset EV3 and for functional clustering Appendix Fig S7. Difference between knockdown and control with *P*-value = 0.049 (Student's *t*-test) is indicated with an asterisk.

B   Kaplan–Meier plot showing that LPS induced a higher mortality rate of wa3 mice than of littermate controls following 2 weekly subsequent injections of LPS (1 mg/kg).

expression was higher in the wa3 hearts were associated with metabolism, signal transduction, fibrosis, and immune function (clustered by GOrilla and David analyses in Appendix Fig S3), pathways that are known to be involved in inflammation and repair (Ooi *et al*, 2015; Prabhu & Frangogiannis, 2016). The immune function-associated genes encode cytokines, cytokine receptors, chemokines, chemokine receptors, metalloproteinases, bactericidal mediators, regulators of T cells, and components of antigen presentation by antigen-presenting cells (MHC class II; Dataset EV1). Further analyzing the transcriptomics data using Digital Cell Quantification (DCQ; Altboum *et al*, 2014; Frishberg *et al*, 2016) to predict *in vivo* relative presence of immune cells in the heart suggest that the 7-week-old wa3 hearts contain more infiltrating CD8[+] T cells and macrophages than controls (Appendix Fig S4). In 12-week-old hearts, 4,992 genes

were differentially expressed (Deseq algorithm "R" *P* < 0.05 threshold). A total of 2,182 genes were twofold to ~120-fold upregulated (Fig 6B and C, and Dataset EV2), and many of them were associated with pathways related to activation of the immune response such as cytokines–cytokine receptors, chemokine–chemokine receptors, Toll-like receptors (TLRs), complement, phagocytosis, cell adhesion, T-cell function and proliferation, TGFβ signaling, and NF-κB signaling (Dataset EV2, and clustering by GOrilla and David analyses in Appendix Fig S5). The computational DCQ analysis predicted a higher presence of γδT cells and dendritic cells in the 12-week-old wa3 hearts than controls (Appendix Fig S6). γδT cells are involved in many chronic inflammations, such as inflammatory DCM (Takeda *et al*, 2008; Li *et al*, 2014; Fay *et al*, 2016). To further study *in vivo* the cardiac response to inflammatory triggers in the absence

of *Ppp1r13l,* LPS was injected to 12-week-old wa3 mice and littermate controls. Two hours following LPS injection, the expression of *Il1b* mRNA, as well as of *Il6,* was further increased in wa3 hearts compared to WT hearts (Fig 7A). The upregulation of *Tnfa* may require longer stimulation. The mRNA expression levels of selected DCM-associated genes were regulated similarly in wa3 and WT mice (Fig 7A). RNA-seq analysis indicated a higher inducible expression of genes associated with acute inflammation in the hearts of LPS-injected wa3 mice than in LPS-injected controls (Dataset EV3, and clustering by GOrilla and David analyses in Appendix Fig S7). In accordance, two consecutive weekly injections of low amount of LPS demonstrated a lower resistance of wa3 mice in response to the inflammatory stimulus (Fig 7B); while 8 out of 9 LPS-injected wa3 mice died during the first 2 weeks of the experiment, none of the LPS-injected WT or non-injected wa3 mice died.

## Discussion

This study presents a novel form of human CCS associated with a premature stop codon in *PPP1R13L*. The phenotype is characterized by early onset of severe progressive DCM, with predominance of either the left or right ventricle, progressing into a fatal state within the first 3 years of life. Skin and hair manifestations are mild and may include sparse, dry, and woolly hair, ichthyosis-like fine scale and erythema, wedge-shaped teeth, and dystrophic nails. There is no evidence for palmo-plantar keratoderma, suggestive of non-hidrotic ectodermal dysplasia, and thermoregulation is normal. These skin manifestations together with severe DCM define a novel CCS.

Although we have studied only one SeVa that affects the function of *PPP1R13L* in four families, and most probably related to a single founder, the data strongly indicate a causative role for this gene and SeVa in the patients' CCS. Firstly, mice that carry spontaneous AR mutation in *Ppp1r13l* possess a similar phenotype (Herron *et al*, 2005; Toonen *et al*, 2012). Cardiomyopathy and woolly hair were also reported in cattle that carry a frameshift SeVa in the bovine ortholog gene (Simpson *et al*, 2009). Secondly, detection of myocarditis in patient VI$_{10}$ motivated us to look for the function of iASPP in cardiac inflammation; our results demonstrated that iASPP is required for lifting the cardiac threshold response to inflammatory triggers. The idea that over-expression of typical pro-inflammatory mediators such as the cytokines IL-1B, TNFα, and IL-6, as we found in the absence of iASPP, leads to DCM is in accordance with established data showing that the presence of these mediators in plasma, as well as in the myocardium itself, potentiates cardiac remodeling processes such as hypertrophy, ventricular dilation, fibrosis, and apoptosis (Gullestad *et al*, 2012; Dick & Epelman, 2016; Prabhu & Frangogiannis, 2016).

Cardiomyocytes express the LPS detector TLR4, and its deficiency protects against LPS-induced cardiac dysfunction and remodeling (Mann, 2002; Nemoto *et al*, 2002; Shimamoto *et al*, 2006; Arslan *et al*, 2010). Cardiomyocytes also express TLR3 that can be activated by viral RNA (Yajima & Knowlton, 2009), and other pattern recognition receptors as the receptor for advanced glycation end product (RAGE) all are well known as NF-κB activators. NF-κB is a central transcriptional effector of inflammatory signaling. The inducible activity of NF-κB is initially cardio-protective, since early inflammatory functions are necessary for the transition to later proper physiological reparative program, but prolonged activation can lead to sustained tissue damage and improper healing, defective scar formation, and heightened cell lose and contractile dysfunction (Gordon *et al*, 2011; Prabhu & Frangogiannis, 2016). NF-κB activation triggers transcription of a large repertoire of genes including inflammatory cytokines, chemokines, and adhesion molecules (Gordon *et al*, 2011; Prabhu & Frangogiannis, 2016). These inflammatory mediators, in turn, further amplify the inflammatory response by attracting immune cells. NF-κB inhibition during myocardial ischemia/reperfusion decreases infract size, reduces inflammatory responses including leukocyte infiltration, and improves cardiac function. (Morishita *et al*, 1997; Xuan *et al*, 1999; Squadrito *et al*, 2003; Onai *et al*, 2004; Brown *et al*, 2005; Li *et al*, 2007; Moss *et al*, 2007; Tranter *et al*, 2010; Zhang *et al*, 2013). Silencing of iASPP increased the binding activity of NF-κB to promoters of pro-inflammatory cytokines in human fibroblasts, and NF-κB inhibition blocked the hyper-responsiveness of these cells to LPS. In general, the inducible transcriptional programs in the absence of iASPP contain many classic pro-inflammatory targets of NF-κB. Further study is required to dissect the mechanism underlying the ability of iASPP to inhibit the DNA-binding activity of NF-κB.

*Ppp1r13l* knockdown in murine cardiomyocytes promoted a wide transcriptional program associated with innate immune response. Moreover, the absence of iASPP in wa3 mice demonstrated a dynamic *in vivo* development of inflammatory conditions, starting with differential expression of the *s100a9* in newborns. The plasma level of S100A9/S100A8 appears to be a marker of cardiovascular risk in human. *s100a9*$^{-/-}$ mice are largely protected from endotoxin-induced cardiomyocyte dysfunction, measured as reduced ejection fraction, probably as a result of altered calcium flux following signaling through TLR4 and RAGE, indicating a pro-inflammatory role. However, anti-inflammatory functions have also been described. From that time point, a gradual increase in the presence of pro-inflammatory mediators was observed in 7-week, and then in 12-week-old wa3 mice, and further following LPS injection. Collectively, these results indicate a critical role for iASPP in maintaining normal heart response to environmental triggers of inflammation, at least partially by inhibition of NF-κB activity. In the absence of iASPP, the normal heart physiology and stress response is fatally disturbed.

Apart from NF-κB, iASPP interacts with other transcription factors such as p53 (Chikh *et al*, 2011; Cai *et al*, 2012; Kramer *et al*, 2015; see Appendix Figs S8–S10) and p53 family members p63 and p73 (Chikh *et al*, 2011; Notari *et al*, 2011). Perhaps, the interaction of iASPP with other transcription factors contributes to the complex CCS phenotype. For example, it was shown that iASPP is involved in maintenance of epidermal homeostasis in mice (Chikh *et al*, 2011, 2014; Notari *et al*, 2011), partially by participating in p63-mediated regulation of gene expression (Chikh *et al*, 2011). p63 is also involved in heart development (Paris *et al*, 2012), and iASPP may exert some of its activities in cardiomyocytes by modulating its function. In addition, murine iASPP was recently reported in association with desmosomes in cardiomyocytes (Notari *et al*, 2015). Mutations in desmosomal encoding genes are underlying other CCSs such as the Naxos and Carvajal, in which cardiac disease becomes symptomatic during adolescence (Bolling & Jonkman, 2009; Rickelt & Pieperhoff, 2012), although not in infancy.

Mice with mutations in *Ppp1r13l* are born with abnormal open eyelids (Herron *et al*, 2005; Toonen *et al*, 2012). Closed eyelids in the first days serve as a protective barrier preventing premature exposure of the developing ocular structures to the environment. The eyes in adult *Ppp1r13l*-mutated mice are identified with severe corneal opacities, abnormalities of the anterior eye segment, and absence of meibomian glands (Toonen *et al*, 2012). Patient $VI_4$ presents a similar phenotype. Perhaps, the ocular abnormalities of Patient $VI_4$ resulted from absence of iASPP, and this effect is partially penetrant as it was not observed in the other four patients. This observed similarity between *Ppp1r13l* mutated mice and our patient needs further exploration.

In summary, we identified *PPP1R13L* as a novel gene underlying human AR-CCS with fatal DCM, and presume its absence affects a subset of familial and sporadic inflammation-associated DCMs. Our data indicates involvement of iASPP in the transcriptional regulation of genes necessary for normal balanced response to inflammatory triggers. We suggest a scenario in which iASPP is a rheostat that uplifts the bar for innate immune response of cardiomyocytes and possibly of other cells in the heart such as fibroblasts. The lowered threshold in cardiomyocytes with SeVa that affects the function of *PPP1R13L* promotes recurrent and prolonged inflammatory processes, in possibly already predisposed heart, and ultimately adverse repair and overactive fibrosis that consequently lead to the severe DCM during infancy. Our data increase the repertoire of genes underlying predisposition to heart remodeling, failure, and possibly sudden death following heart infections or other deleterious processes. Although experimental work has established a crucial role for the inflammatory cascade in cardiac repair and remodeling, to date, there has been no successful clinical immunomodulatory or anti-inflammatory therapeutic strategies for heart diseases (Prabhu & Frangogiannis, 2016). Since the reparative response is a highly dynamic process, as can be learned from the alteration in the transcriptional patterns during diseases development, a successful immunomodulatory therapy should require careful design implying knowledge on the time course of the inflammatory process and the temporary associated immune cells. The pathophysiologic heterogeneity based on the genetic background of patients with heart diseases has also important therapeutic implications, for example, patients with impaired iASPP function may be predisposed to dilative remodeling after myocarditis or myocardial infraction and therefore benefit from personalized tailored anti-inflammatory intervention. Altogether, this study may lead to the development of new strategies for early diagnosis of individuals and populations at risk, accurate genetic counseling, prenatal diagnosis, and hopefully novel treatment modalities.

# Materials and Methods

### Ethic statement

Patients and families—The IRB of Galilee Medical Center and the Israeli Ministry of Health approved the study. All study participants and parents of minors signed informed consent forms. The murine studies have been reviewed and approved by the Inspection Committee on the Constitution of Animal Experimentation at Bar-Ilan University.

### Patients and families

Five infants presenting with CCS were ascertained and included in this study (Fig 1A). Family history was taken, and physical, cardiologic, dermatologic, and ophthalmologic examinations were conducted, including imaging of the heart by conventional echocardiogram and transthoracic echocardiogram (TTE), CT, and Holter ECG. Laboratory tests, metabolic measurements including muscle biopsy for enzymatic activity of the mitochondrial respiratory chain complexes, were performed. Blood was drawn for molecular studies and sequencing of suspected causative genes. Fibroblast cell cultures were established from skin biopsies.

### Mice

Female BALB/c mice were purchased from Harlan Biotech, Israel, and maintained under pathogen-free conditions. Female Wa3 mice (BALB/cJ-*Ppp1r13l*$^{wa3}$/J) were purchased from The Jaxon Laboratory. These mice carry deletion and insertion mutations in exon 12 of the *Ppp1r13l* gene that generate premature stop codon. The mice were raised under SPF conditions in the animal facility of The Faculty of Medicine in Safed. All experiments were approved by the ethics committee of Bar-Ilan University, Ethics number 5-01-2014. Total number of mice participated and sacrificed in this research is 140. Mice were chosen randomly out of genotyped pool of mice in the same age (0, 7, or 12 weeks).

### Whole exome deep sequencing (WES) and data analyses

WES (100X) was performed (Atlas Biolabs, Germany). Genomic DNA libraries were created following standard protocols for high-throughput paired-end sequencing. Libraries were enriched using Roche Nimblegen v2 Exome enrichment kit for known coding loci in the human genome (44.1 Mbp; Roche Nimblegen, Indianapolis, IN) and sequenced on Illumina HiSeq2000 at $> 100\times$ coverage ($2 \times 100$ bp PE run; Illumina Inc., San Diego, CA). Base calling, quality score assessment, and variant calling were completed using a combination of available tools and custom scripts, as previously described (Paris *et al*, 2012). Variant effects were determined using snpEff 3.0 (Cingolani *et al*, 2012). Protein changing variants were filtered to private and rare (minor allele frequency < 0.01) presumptively damaging variants. Sanger sequencing and RFLP analyses were conducted as previously described (Falik Zaccai *et al*, 2014).

### SNP analysis and homozygosity mapping

Whole genome SNP analysis was performed on genomic DNA of the patient $VI_{10}$, using an Illumina Human 1M–Duo DNA Analysis BeadChip and Illumina's standard hybridization protocol. The data were analyzed using the GenomeStudio software (Illumina, San Diego, CA). Regions of homozygosity were identified based on "B allele frequency" plots.

### Mutation validation and co-segregation

Sanger sequencing and RFLP analyses were conducted as previously described (Falik Zaccai *et al*, 2014). Genomic DNA was amplified by PCR, using primers flanking exon 11: 5′-GCAGCCTCTCACGCA

TCGTT-3′ forward primer and 5′-CCATGTCCCCTTTTCCCTTA-3′ reverse primer. The mutation abolishes a restriction site for BstNI enzyme (New England Biolabs, Ipswich, Massachusetts, USA).

### Histology of the heart

Post-mortem immunohistochemical hematoxylin and eosin staining of cardiac tissue were performed according to standard staining procedures. Images were obtained via Olympus BX-51 digital microscope (Olympus America) equipped with an UPlanFL 2X/0.50 numeric aperture objective and Olympus DP70 digital camera system.

### Human fibroblast cultures

Patient skin-derived fibroblasts (PDFs) and control age-matched skin-derived fibroblasts (CFs) cultured in high-glucose (4.5 g/l) DMEM supplemented with 15% FBS, 2 mM L-glutamine, MEM non-essential amino acid solution, and penicillin–streptomycin.

### Western blot analysis

Total protein lysate or cytosolic/nuclear fractions were extracted and separated by SDS polyacrylamide gel electrophoresis, transferred to PVDF membranes, and reacted with the following antibodies: anti-Rel-A (bethyl, A301-824A), anti-iASPP (Santa Cruz, H-300, sc-98538), anti-Lamin-B (Santa Cruz, m-20, sc-6217), and anti-β-actin (Santa Cruz, N-21) all used at the concentration of 1 μg/ml.

### Chromatin immunoprecipitation (ChIP) assay

ChIP assay was done as previously described (Jacob et al, 2011). Briefly, fibroblasts were cross-linked in 1% formaldehyde for 10 min at RT followed by glycine quenching. Cells were then washed and lysed, and the chromatin was sonicated into 200- to 1,000-bp fragments, diluted, and pre-cleared using protein A beads. Anti-Rel-A- (Santa Cruz) or isotype-matched IgG (control) antibodies were incubated with the chromatin samples overnight. Protein A beads were added for 2 additional hours, and the beads were then washed and the cross-linked reversed overnight at 65°C. The DNA was purified and subjected to quantitative PCR analysis using the indicated primers. The dissociation curves following amplification showed that all primer pairs generated single products. The amount of PCR product amplified was calculated relative to the input.

Primers:
hTNFA promoter: For: CCTCCAGATGAGCTCATGGGTT, Rev: GGG TGTGCCAACAACTGCCTTT.
hIL1B promoter: For: TGTCTTCCACTTTGTCCCACA, Rev: CGTT GTGCAGTTGATGTCCA.
hp21 promoter: Validated primer set was purchased from Millipore.
hIKB promoter: Validated primer set were purchased from Millipore.

### Isolation of murine neonatal cardiomyocytes

Mouse neonatal cardiomyocytes were isolated using Worthington Rat neonatal isolation kit. Briefly, hearts were isolated from 1- to 4-day-old BALB/c mice, minced, and incubated overnight at 4°C.

Trypsin inhibitor was added, and heart pieces were incubated for 35 min in the presence of collagenase at 37°C. Cells were passed through 100-μm cell strainer, fibroblasts were removed by adhesion for 2 h, and non-adhered cells were centrifuged (50 g/5 min) and seeded at concentration of $3 \times 10^4$ cells/well in gelatin-coated 6-well plates in a myocytes growth medium (promocell) for 5 days.

### Measurement of mRNA expression

Fibroblasts or cardiomyocytes were seeded overnight and left unstimulated or stimulated with 0.5 μg/ml LPS (Sigma) for the indicated time points. RNA was prepared for qPCR using Fast sybr green master mix (Applied Biosystems). Results were normalized to GAPDH and are the mean of at least three independent experiments. SD differences with $P$-values $\leq 0.05$ (Student's $t$-test) are indicated with an asterisk. The primers that were used are as follows:

Human:
PPP1R13L For: GGCGGTGAAGGAGATGAAC Rev: AGATGAGGA AATCCACGATAGAGA, TNFA For: GAGGCCAAGCCCTGGTATG Rev: CGGGCCGATTGATCTCAGC, IL6 For: CCTGAACCTTCCAAAG ATGGC Rev: TTCACCAGGCAAGTCTCCTCA, IL1B For: ATGATG GCTTATTACAGTGGCAA Rev: GTCGGAGATTCGTAGCTGGA, IL8 For: CTGCGCCAACACAGAAATTATTGTA Rev: TTCACTGGCAT CTTCACTGATTCTT, MCP1 For: ACTCTCGCCTCCAGCATGAA Rev: TTGATTGCATCTGGCTGAGC, MMP9 For: TGTACCGCTATGGTTA CACTCG Rev: GGCAGGGACAGTTGCTTCT, IKB For: GTCTTTGCAC ATCATTCGTGGG Rev: GTGCCGAAGCTCCAGTAGTC, GAPDH For: GTCAAGGCTGAGAACGGGAA Rev: AAATGAGCCCCAGCCTTCTC.

Mouse:
Ppp1r13l For: TAGAGGCCCGTTTTGGACG Rev: CCCGATCTAG GCTGCTGTAG, Il1b For: GGGCCTCAAAGGAAAGAATC Rev: TTCTTTGGGTATTGCTTGGG, Il6 For: TCACTTTGAGATCTAC TCGGCAAACC Rev: TCTGACCACAGTGAGGAATGTCCA, Tnfa For: GATTATGGCTCAGGGTCCAA Rev: ACAGTCCAGGTCACTGTCCC, CCCCTCATTCCTTACCACCC, Gapdh For: CTCCCACTCTTCCACC TTCG Rev: CCACCACCCTGTTGCTGTAG.

### RNA interference

Lentiviral particles expressing PPP1R13L-shRNA or control scrambled-shRNA (MISSION-tGFP, Sigma) were produced in HEK-293T cells, kindly provided by Gal-Tanami Lab and checked to be mycoplasma free. The supernatants were collected 24 h post-transfection for 8 h and used immediately for transduction in the presence of polybrene (8 μg/ml) by centrifugation for 1 h at 863 g (32°C). The following shRNA sequences were used:

mPpp1r13l-shRNA1
CCGGCCCAGACTGAAGATTAGGAAACTCGAGTTTCCTAATCTTCA GTCTGGGTTTTTG(TRCN0000087798, Sigma)
mPpp1r13l-shRNA2
CCGGGCGCAACTACTTCGGGCTCTTCTCGAGAAGAGCCCGAAGTA GTTGCGCTTTTTG(TRCN0000087801, Sigma)
hPPP1R13L-shRNA
CCGGCCAACTACTCTATCGTGGATTCTCGAGAATCCACGATAGAG TAGTTGGTTTTTT(TRCN0000022209, Sigma)

### *Ppp1r13l* silencing

Neonatal cardiomyocytes were seeded in gelatin-coated plates and transduced with *Ppp1r13l*-shRNAs or control non-targeting-shRNA, both express the reporter GFP. On the fifth day, the cells were exposed to 0.5 μg/ml LPS or left unstimulated.

### *Ppp1r13l*-deficient hearts

Hearts were isolated following perfusion and homogenized in lysis buffer (Norgen Biotek) for RNA extraction.

### RNA sequencing (RNA-Seq)

Total RNA was extracted using RNA extraction kit (Norgen). Poly-A mRNA was further purified using Poly(A) mRNA Magnetic Isolation Module (NEBNext®), and RNA concentration was determined using high-sensitivity Qbit® RNA kit (molecular probes). RNA library was constructed using Ultra™ RNA library prep kit for Illumina® (NEBNext®) following manufacturer instructions. Size and quality of samples were examined by bioanalyzer (Agilent Technologies), and 50-bp single read was sequenced. The data were mapped using bowtie and further analyzed for differential expression with Seqmonk (http://www.bioinformatics.babraham.ac.uk/projects/seq monk/).

### Bioinformatics

RNA-seq data were first aligned to the mouse transcriptome, Ensembl mouse gene build mm9 using Bowtie (Langmead *et al*, 2009), mmseq (Turro *et al*, 2011) and htseq-count (Anders *et al*, 2015) were used to assign reads to genes. DESeq package (Anders & Huber, 2010) was then used to normalize read count to library size. Additionally, non-expressed genes were removed when the number of reads of at least one replicate had at least 10 reads. For each replicate, the reads of time t1 and t2 of the simulated were normalized to time t0 of the corresponding treatment. Clustering was performed online by DAVID (Huang da *et al*, 2009a,b) and GOrilla analyses (Eden *et al*, 2009). Digital Cell Quantification (DCQ) analysis was performed according to the algorithm described in Altboum *et al* (2014) and Frishberg *et al* (2016). This computational method combines genome-wide gene expression data with a mouse immune cell compendium has already been used to predict immune cells populations during influenza virus infection (Leist *et al*, 2016; Tisoncik-Go *et al*, 2016).

### Statistics

Figures 2A, 3, 4B and C, 5B and 7A: qPCR experiments were conducted at least three times (biological repeats) with technical triplicates. Statistical significance was calculated using *t*-test with the appropriate assumptions for each experiment (one-tailed, equal variance) after variance calculation (Stdev only).

Figure 6B and C: RNA sequencing: All statistical calculation and analysis were conducted using Seqmonk free software (http://www.bioinformatics.babraham.ac.uk/projects/seqmonk/) using ready to use codes of "R" algorithms for the calculation of differentially expresses genes with *P*-value < 0.05.

---

**The paper explained**

**Problem**

Dilated cardiomyopathy (DCM) is a life-threatening disorder and is currently responsible for ~10,000 deaths and 46,000 hospitalizations each year in the United States. Five Arab Christian–Israeli infants were diagnosed with fatal DCM associated with mild skin, teeth, and hair abnormalities. All passed away before age 3.

**Results**

*PPP1R13L* encoding the iASPP protein was identified as the novel gene underlying this human cardio-cutaneous syndrome. iASPP was found as a regulator of transcriptional programs associated with normal cardiac response to inflammatory triggers. Its absence increased the inflammatory response of the heart as soon as with birth, and more dramatically with age and after exposure to inflammatory stimuli.

**Impact**

Our study determined *PPP1R13L* as the gene underlying a novel autosomal-recessive cardio-cutaneous syndrome in humans and strongly suggests that the fatal DCM during infancy is a consequence of failure to regulate transcriptional pathways necessary for tuning cardiac threshold response to common inflammatory stressors. Patients with impaired iASPP function may be expected to benefit from personalized tailored anti-inflammatory intervention. This study may lead to the development of new strategies for early diagnosis of individuals and populations at risk, accurate genetic counseling, prenatal diagnosis, and, hopefully, novel treatment modalities.

---

Expanded View for this article is available online.

## Acknowledgements

We thank the families who participated in this study and the physicians and nurses who helped in the care of these patients; Irit Gat-Viks laboratory, Cell Research and Immunology Department, Tel Aviv University, Tel Aviv, Israel, for help with digital cell quantification; and Tobie Kuritsky for her assistance in preparing this manuscript for submission. Research was supported by grants from the Israel Science Foundation (OA), Gessner Fund for Medical Research in Cardiology (OA and TF-Z) and "Izvonot" Foundation of the Israeli Ministry of Justice, Jerusalem, Israel (TF-Z).

## Author contributions

TCF-Z initiated the project, recruited the patients and families, supervised the human clinical and molecular genetic studies, designed the study, and wrote the manuscript. YB performed molecular and biochemical experiments of Figs 2, 3, 4, 5, 6 and 7, and associated supplementary data and participated in preparing figures, tables, and writing the manuscript. HM performed clinical and metabolic workup of the patients. MS performed molecular and biochemical experiments of Figs 2, 3, 4 and 5, and associated supplementary data and participated in preparing figures. SG performed molecular and *in vivo* experiments of Figs 6 and 7, and associated supplementary data and participated in preparing figures. AL performed cardiologic workup and follow-up on two patients. LK performed sequencing studies of the patients, tissue cultures, and expression analyses and participated in preparing figures, tables, and writing the manuscript. SBH performed clinical workup and follow-ups of the patients. AS ascertained and performed clinical workup and follow-up the third patient. LG-Y performed the cardiologic workup of the third patient. SC performed the genetic counseling to the families presented in this work. DRS performed the

molecular analyses of the patients and performed sequencing studies and segregation analyses. MK participated in the population screening. MW participated in molecular experiments of Fig 2. IL participated in the expression studies of *PPP1R13L* in Fig 1. YS participated in the molecular and bioinformatic analyses in Fig 1. GT participated in the metabolic workup. SS participated in the population screening. ER participated in the bioinformatics analysis of the RNA-seq. EA-H performed the dermatological workup of the patients and sequencing of the cardio-cutaneous-related genes. EV performed the pathological workup on the patients. LA-S performed the pathological workup on the patients. DG performed the five radiological workup of the patients. RB performed the dermatological workup of the patients and sequencing of the cardio-cutaneous-related genes. ZS performed the ophthalmological workup of the patients. OB-D participated in establishing fibroblasts tissue cultures of the patients. OA conceived and designed the study, supervised the molecular and biochemical experiments, and wrote the manuscript.

## Conflict of interest

The authors declare that they have no conflict of interest.

## For more information

http://www.uptodate.com/contents/causes-of-dilated-cardiomyopathy?source = search_result&search = Causes + of + dilated + cardiomyopathy&selectedTitle = 1 ~ 150

http://www.uptodate.com/contents/genetics-of-dilated-cardiomyopathy?source = search_result&search = Genetics + of + dilated + cardiomyopathy&selectedTitle = 1 ~ 150

http://www.bioinformatics.babraham.ac.uk/projects/seqmonk/

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
