## [Review Process File · EMBO Molecular Medicine]

Sequence variation in PPP1R13L results in a novel form of cardio-cutaneous syndrome

Tzipora C. Falik-Zaccai, Yiftah Barsheshet, Hanna Mandel, Meital Segev, Avraham Lorber, Shachaf Gelberg, Limor Kalfon, Shani Ben Haroush, Adel Shalata, Liat Gelernter-Yaniv, Sarah Chaim, Dorith Raviv Shay, Morad Khayat, Michal Werbner, Inbar Levi, Yishay Shoval, Galit Tal, Stavit Shalev, Eli Reuveni, Emily Avitan-Hersh, Eugene Vlodayvsky, Liat.Appl-Sarid, Dorit Goldsher, Reuven Bergman, Zvi Segal, Ora Bitterman-Deutsch, and Orly Avni

Corresponding authors: Tzipora Falik-Zaccai and Orly Avni, Galilee Medical Center

Review timeline:	Submission date:	31 March 2016
	Editorial Decision:	01 June 2016
	Revision received:	10 October 2016
	Editorial Decision:	09 November 2016
	Revision received:	05 December 2016
	Accepted:	12 December 2016

Transaction Report:

Editor: Roberto Buccione

1st Editorial Decision

01 June 2016

Thank you for the submission of your manuscript to EMBO Molecular Medicine. We have now heard back from the three Reviewers whom we asked to evaluate your manuscript.

We are sorry that it has taken longer than usual to get back to you on your manuscript. In this case, we experienced significant difficulties in securing three expert and willing Reviewers. Further to this the evaluations were delivered with some delay and, finally, the decision required further discussion.

As you will see, although Reviewers 1 and 3 are more appreciative of your study, all three raise significant and fundamental issues that in aggregate, I am afraid, preclude publication of the manuscript in EMBO Molecular Medicine at this time. I will not discuss each point in detail as they are clearly stated.

The main issues, notwithstanding different perceptions on the overall interest of the study, include 1) the lack of clear casual evidence linking the in vitro observations to the in vivo phenotype, and 2) the need for further experimentation (including on the mouse models) to provide sufficient mechanistic insight, especially in terms of the how the pro-inflammatory phenotype is causing fatal DCM.

Given these fundamental concerns and the general lack of enthusiasm by the Reviewers, I have no

choice but to return the manuscript to you at this stage. I'm afraid that we agree with their opinions and in our assessment it is not realistic to expect to be able to address these issues experimentally and to the satisfaction of the Reviewers in a reasonable time frame.

I wish to add however that, considered the potential interest of these findings, we would have no objection to consider a new manuscript on the same topic if at some time in the near future you have obtained data that would considerably strengthen the message of the study and address the Reviewers' concerns.

I am sorry to have to disappoint you at this stage and again apologise for the inevitable delay. I hope that the Reviewers' comments will be helpful in your continued work in this area.

***** Reviewer's comments *****

Referee #1 (Comments on Novelty/Model System):

The *ppp1r13l* knockout mice are known to exhibit wavy hair and progressive dilated cardiomyopathy (DCM) phenotypes. Falik-Zaccai et al found genetic variation in *PPP1R13L* can cause a novel form of cardio-cutaneous syndrome in human. This is interesting findings, and the genetic evidence is strong.

However, I found the mechanistic links are weak: how pro-inflammatory leads to dys-regulation of DCM related genes? The authors should provide mechanistic insights how *PPP1R13L* affects NF- κ B signaling and subsequent expression of DCM related genes.

Referee #2 (Remarks):

This manuscript is a case report of several children who developed DCM and also harbor a loss of function of the *PPP1R13L* gene. The data primarily make the case for an association of the genetic variation and the phenotype, and causality cannot be established based on the studies presented. The strongest evidence is the phenotype of KO mice which have a KO of the *PPP1R13L* gene, which resembles features found in the affected cases. Knockdown studies in cardiac fibroblasts and neonatal cardiomyocytes suggest that effect on inflammatory signaling pathways might underlie the onset of DCM. However, there is no direct evidence that the onset of the DCM phenotype in vivo relates to these in vitro findings and studies in the KO mouse model would strengthen this claim, both at the molecular pathway level, and in terms of blocking the onset of DCM.

Without these data, the study is primarily a case study with a few additional suggestive results that might infer pathways that link genotype and phenotype. Taken together, my view is that it should be published somewhere but probably better for a specialty journal like European Heart Journal.

Referee #3 (Remarks):

In recent years an increasing number of genetic risk factors for cardiomyopathies have been determined, thereby elucidating the pathogenesis of previously idiopathic cardiomyopathies. In this report the authors present strong and novel evidence that a severe childhood cardiomyopathy in consanguineous patients is caused by defective gene *PPP1R13L*. In addition to an inflammatory dilated cardiomyopathy (DCM), the patients present with distinct skin and eye phenotypes, thereby classifying this disorder as cardio-cutaneous syndrome (CCS). The same combination of symptoms has been previously described in *PPP1R13L* deficient mice. *PPP1R13L* is known to regulate/suppress aspects of NF κ B-mediated inflammatory signaling. The present work presents novel evidence that a *PPP1R13L* defect (or shRNA knockdown) indeed leads to an enhanced inflammatory cytokine expression in the CCS context, i.e. in patient-derived fibroblasts or mouse cardiomyocytes/hearts. The enhanced expression of inflammatory mediators is especially enhanced upon challenge of cells by inflammatory stimuli, i.e. LPS. These data are highly suggestive, however, formal proof is lacking that this hyper-inflammation is indeed causing the deadly DCM in the patients. This is my major concern about this work, which could be addressed experimentally in

the PPP1R13L deficient mouse. As described by Herron et al (2005) these mice develop DCM, however, the mice are able to live at least 8 months. Would a low dose LPS challenge of younger mice (i.e. 6 wk old) trigger the development of DCM as compared to PPP1R13L deficient mice kept in the fairly clean environment of an animal facility? Such an experiment would not only elucidate the pathogenesis of DCM but would also have clinical impact on management of such patients by suggesting an anti-inflammatory/anti-bacterial strategy. Please discuss such strategies.

Besides this major comment I have a few minor points:

- 1) Figure legends do not match the panels presented, i.e. Figure 1.
- 2) Figure 1 is very complex and should be broken into more figures and more clearly presented.
- 3) Figure 5 A presents mouse / heart pictures that have been reported more convincingly and in more detail before (Herron 2005). I suggest omitting those pictures and focus on inflammation and DCM as in my major comment.
- 4) page 6/line 2: What is "AR CCS" ?

1st Revision - authors' response

10 October 2016

Thank you for the insightful review of our manuscript, and for your offer to reconsider a new version of our manuscript on the same topic.

In this new version of the manuscript we present data that strengthens the main point of the manuscript, i.e.- that absence of *PPP1R13L* results in hyper-sensitivity to inflammatory triggers, and as a consequence, leads to fatal dilated cardiomyopathy (DCM).

Our study was appreciated by the reviewers, but they had some concerns that were summarized by you as follows:

1. the lack of clear casual evidence linking the *in vitro* observations to the *in vivo* phenotype
2. the need for further experimentation (including on the mouse models) to provide sufficient mechanistic insight, especially in terms of how the pro-inflammatory phenotype is causing fatal DCM.

We would like to address these concerns point by point.

1. "The lack of clear casual evidence linking the *in vitro* observations to the *in vivo* phenotype"

In the previous version we showed that patients' derived fibroblast and *ppp1r13l*-knocked down cardiomyocytes demonstrated significant hyper-sensitivity to inflammatory stimulus. In this new version of the manuscript we clearly link this phenomenon to the cardiac phenotype of the murine model, carrying spontaneous mutation in *ppp1r13l* (wa3 mice). The new data, which are presented in a new section in the Results entitled: "*Dynamic development of inflammatory transcriptional patterns in ppp1r13l-deficient hearts*" (including new Figures 6-7, Figures 3S-7S, and Tables 1S-3S), demonstrate *in vivo* dynamic alterations in cardiac transcriptional programs in the absence of *ppp1r13l* during DCM development. In this section, by using RNA-seq for newborn, 7 week- and 12-week-old wa3-derived hearts, we clearly show the development of inflammatory conditions, starting with differential expression of only one gene; the *s100a9* in newborns. *S100a9* is known to be associated with heart inflammation (Averill, Kerkhoff et al., 2012). From that point, we observed a gradual (less in 7-week old mice and much more in 12-week old mice) wa3-selective increase in the expression of pro-inflammatory mediators, such as those associated with cytokines-cytokine receptors, chemokine-chemokine receptors, TOLL-like receptors (TLRs), complement, phagocytosis, cell adhesion, T cell function and proliferation, TGF β signaling, and NF- κ B signaling pathways. To further study *in vivo* the cardiac response to inflammatory triggers in the absence of *Ppp1r13l*, LPS was injected to 12-week-old wa3 mice, and the results of the RNA-seq analysis indicated a higher inducible expression than in controls of genes associated with acute inflammation. In accordance, two consecutive weekly injections of low amount of LPS demonstrated a lower resistance of wa3 mice in response to the inflammatory stimulus; almost all of the mutants died during these two weeks while none of the controls. Although our results cannot exclude predisposition of the wa3 hearts to inflammatory triggers, all together our data indicate a crucial requirement for *ppp1r13l* in lifting the cardiac threshold response to inflammatory triggers. In its absence, low doses of inflammatory stimuli are fatal.

2. "The need for further experimentation (including on the mouse models) to provide sufficient mechanistic insight, especially in terms of the how the pro-inflammatory phenotype is causing fatal DCM"

The involvement of inflammatory cytokines in DCM promotion is quite established for many years, although the exact molecular mechanisms underlying this causative effect are not completely understood. We mentioned it in the second paragraph of the Discussion: "The idea that over-expression of typical pro-inflammatory mediators such as the cytokines IL-1B, TNF α , and IL-6, as we found in the absence of iASPP, leads to DCM is in accordance with established data showing that the presence of these mediators in plasma, as well as in the myocardium itself, potentiate cardiac remodeling processes such as hypertrophy, ventricular dilation, fibrosis and apoptosis (Dick & Epelman, 2016, Gullestad, Ueland et al., 2012, Prabhu & Frangogiannis, 2016)". Actually, expression of pro-inflammatory mediators and the inducible activity of NF- κ B downstream are initially cardio-protective, since early inflammatory functions are necessary for the transition to later proper physiological reparative program, but prolonged activation can lead to sustained tissue damage and improper healing, defective scar formation, heightened cell loss and contractile dysfunction, which eventually promote DCM. Therefore, our results demonstrating overall increased expression of genes associated with inflammation, fibrosis and tissue repair, are most probably the reason for the early appearance of the fatal DCM (before age 6 month). Our results can promote the development of better diagnostic tools for dissecting the stage of the disease and most importantly new therapeutic approaches, to prevent and/or to alleviate the development of the disease

Referee#1

Referee#1 agreed that we demonstrated interesting findings, and that the genetic evidence is strong. However, Referee#1 found that the mechanistic links are weak: "how pro-inflammatory leads to dysregulation of DCM related genes?" As we explained above the connection between inflammation and consequent fibrosis and deleterious reparative machinery and DCM is well established (Dick & Epelman, 2016, Gullestad et al., 2012, Prabhu & Frangogiannis, 2016), although the exact molecular mechanisms and the sequence of dynamic alterations have not totally been dissected yet, and can be case-specific. Our results contribute insights into the sequential events toward DCM, and may suggest future stage-specific markers for diagnosis, prevention, and case-specific medical intervention.

Referee#1 also thought that we should provide mechanistic insights for how *PPP1R13L* affects NF- κ B signaling and subsequent expression of DCM related genes. Actually, in the previous version we demonstrated that the hyper-sensitivity to inflammatory trigger in the patients' derived fibroblasts was NF- κ B-dependent (Figure 4). In the new version we demonstrate that many of the inflammatory associated pathways, which their expression is higher in wa-3 mice, are well known as NF- κ B target genes (such as cytokines and chemokines). Although previous studies demonstrated an association between iASPP and NF- κ B (Herron, Rao et al., 2005, Takada, Sanda et al., 2002, Yang, Hori et al., 1999), understanding of the exact molecular mechanism of how iASPP represses NF- κ B will probably need further intensive molecular research. A previous suggestion was that the repressive effect is mainly a consequence of binding competition with other ASPP proteins, which are required for the function of NF- κ B (Sullivan & Lu, 2007). With regards to the DCM associated genes that their expression was downregulated in the *ppp1r13l*-silent newborn cardiomyocytes, our *in vivo* results in wa3 mice, demonstrating that the most earlier observed significant change was in the expression of an inflammatory-associated gene *s100a9*, strongly suggests that alteration in the expression of DCM-associated genes might be secondary to the elevation in cytokine gene expression, as previously demonstrated.

Referee#2

Referee#2 claimed "that Knockdown studies in cardiac fibroblasts and neonatal cardiomyocytes suggest that effect on inflammatory signaling pathways might underlie the onset of DCM. However, there is no direct evidence that the onset of the DCM phenotype *in vivo* relates to these *in vitro* findings and studies in the KO mouse model would strengthen this claim, both at the molecular pathway level, and in terms of blocking the onset of DCM".

As we mentioned above, our new data demonstrated the development of inflammatory conditions *in vivo* in the absence and even stronger in response to inflammatory triggers, and that external inflammatory trigger is lethal for wa3 mice.

Referee#3

We were pleased to realize that Referee#3 thinks that "In this report the authors present strong and novel evidence that a severe childhood cardiomyopathy in consanguineous patients is caused by defective gene *PPP1R13L*... The present work presents novel evidence that a *PPP1R13L* defects (or shRNA knockdown) indeed leads to an enhanced inflammatory cytokine expression in the CCS context, i.e. in patient-derived fibroblasts or mouse cardiomyocytes/hearts. The enhanced expression of inflammatory mediators is especially enhanced upon challenge of cells by inflammatory stimuli, i.e. LPS."

However, the major concern of Referee#3 about our work was that "formal proof is lacking that this hyper-inflammation is indeed causing the deadly DCM in the patients... which could be addressed experimentally in the *PPP1R13L* deficient mouse. As described by Herron et al (2005) these mice develop DCM, however, the mice are able to live at least 8 months. Would a low dose of LPS challenge of younger mice (i.e. 7-week old) trigger the development of DCM as compared to *PPP1R13L* deficient mice kept in the fairly clean environment of an animal facility? Such an experiment would not only elucidate the pathogenesis of DCM, but would also have clinical impact on management of such patients by suggesting an anti-inflammatory/anti-bacterial strategy. Please discuss such strategies."

As mentioned above, we fully agree with Referee#3 and therefore performed, with success, the suggested experiment. As can be seen in Figure 7, LPS injections increased the expression of pro-inflammatory mediators more strongly in the hearts of wa3 mice than controls, and two sequential weekly injections of LPS were lethal selectively for wa3 mice.

Regarding discussion of therapy options, we added the following paragraph to the Discussion: "Although experimental work has established a crucial role for the inflammatory cascade in cardiac repair and remodeling, to date, there has been no successful clinical immunomodulatory or anti-inflammatory therapeutic strategies for heart diseases (Prabhu & Frangogiannis, 2016). Since the reparative response is a highly dynamic process, as can be learned from the alteration in the transcriptional patterns during diseases development, a successful immunomodulatory therapy should require careful design implying knowledge on the time course of the inflammatory process and the temporary associated immune cells. The pathophysiologic heterogeneity based on the genetic background of patients with heart diseases has also important therapeutic implications. e.g. patients with impaired iASPP function may be predispose to dilative remodeling after myocarditis or myocardial infarction, and therefore benefit from personalized tailored anti-inflammatory intervention."

Response to the minor points:

1) Figure legends do not match the panels presented, i.e. Figure 1.
Sorry, the Figure legend has been corrected

2) Figure 1 is very complex and should be broken into more figures and more clearly presented.
Figure 1 was split into Figure 1 and Figure 2

3) Figure 5A presents mouse/heart pictures that have been reported more convincingly and in more detail before (Herron 2005). I suggest omitting those pictures and focus on inflammatory and DCM as in my major comment.

We decided to leave this Figure since the murine background between these two manuscripts is different C57bl/6 (Herron 2005) and Balb/C (this manuscript). We moved the Figure to Supplementary Data Figure 2S, and mentioned that these results were presented previously by Herron 2005.

4) page 6/line 2: What is "AR CCS"?
Autosomal Recessive (AR) CCS, mentioned firstly in Results page 6.

Again, we fully understand that we are submitting our manuscript as a NEW SUBMISSION that includes new data that were requested by the reviewers and yourself.

We hope that our work will now be determined suitable for publication in the journal *EMBO Molecular Medicine*.

References:

Averill MM, Kerkhoff C, Bornfeldt KE (2012) S100A8 and S100A9 in cardiovascular biology and disease. *Arteriosclerosis, thrombosis, and vascular biology* 32: 223-9

Dick SA, Epelman S (2016) Chronic Heart Failure and Inflammation: What Do We Really Know? *Circulation research* 119: 159-76

Gullestad L, Ueland T, Vinge LE, Finsen A, Yndestad A, Aukrust P (2012) Inflammatory cytokines in heart failure: mediators and markers. *Cardiology* 122: 23-35

Herron BJ, Rao C, Liu S, Laprade L, Richardson JA, Olivieri E, Semsarian C, Millar SE, Stubbs L, Beier DR (2005) A mutation in NFkB interacting protein 1 results in cardiomyopathy and abnormal skin development in wa3 mice. *Human molecular genetics* 14: 667-77

Prabhu SD, Frangogiannis NG (2016) The Biological Basis for Cardiac Repair After Myocardial Infarction: From Inflammation to Fibrosis. *Circulation research* 119: 91-112

Sullivan A, Lu X (2007) ASPP: a new family of oncogenes and tumour suppressor genes. *British journal of cancer* 96: 196-200

Takada N, Sanda T, Okamoto H, Yang JP, Asamitsu K, Sarol L, Kimura G, Uranishi H, Tetsuka T, Okamoto T (2002) RelA-associated inhibitor blocks transcription of human immunodeficiency virus type 1 by inhibiting NF-kappaB and Sp1 actions. *J Virol* 76: 8019-30

Yang JP, Hori M, Sanda T, Okamoto T (1999) Identification of a novel inhibitor of nuclear factor-kappaB, RelA-associated inhibitor. *J Biol Chem* 274: 15662-70

2nd Editorial Decision

09 November 2016

Thank you for the submission of your revised manuscript to EMBO Molecular Medicine. We have now received the enclosed reports from the referees that were asked to re-assess it. As you will see the reviewers are now globally supportive and I am pleased to inform you that we will be able to accept your manuscript pending the following final editorial amendments.

You may refer to the author guidelines (<http://embomolmed.embopress.org/authorguide>) for details on how to address most of the following. Do not hesitate to contact the editorial office if you need further help.

***** Reviewer's comments *****

Referee #2 (Remarks):

I think the authors have addressed my initial concerns...I am still a bit underwhelmed by their mechanistic data and the links to the in vivo murine model context, but their experimentation does strengthen their case.

Referee #3 (Comments on Novelty/Model System):

This second version of paper is of high quality throughout. The "medium" novelty is only due to the fact that a DCM in the mouse model had been described 10 years ago. Here comes the relevance for people ... and this should fit to EMBO Mol Med.

Referee #3 (Remarks):

well done - no further comments

2nd Revision - authors' response

05 December 2016

Authors made the requested editorial changes.

Corresponding Author Name: Tzipora Falik-Zaccai
Journal Submitted to: EMBO Molecular Medicine
Manuscript Number: EMM-2016-06523